# Five Years Outcomes and Predictors of Events in a Single-Center Cohort of Patients Treated with Bioresorbable Coronary Vascular Scaffolds

**DOI:** 10.3390/jcm9030847

**Published:** 2020-03-20

**Authors:** Remzi Anadol, Annika Mühlenhaus, Ann-Kristin Trieb, Alberto Polimeni, Thomas Münzel, Tommaso Gori

**Affiliations:** 1Kardiologie I, Zentrum für Kardiologie, University Medical Center Mainz and Deutsches Zentrum für Herz und Kreislauf Forschung, Standort Rhein-Main 55131 Mainz, Germanyannika.muehlenhaus@web.de (A.M.); ann-kristin.trieb@gmx.de (A.-K.T.); polimeni@unicz.it (A.P.); tmuenzel@uni-mainz.de (T.M.); 2Division of Cardiology, Department of Medical and Surgical Sciences, "Magna Graecia" University, 88100 Catanzaro, Italy

**Keywords:** restenosis, thrombosis, bioresorbable scaffolds, coronary stent

## Abstract

Introduction: We report outcome data of patients treated with coronary bioresorbable scaffolds up to 5 years and investigate predictors of adverse events. Methods: Consecutive patients treated with at least one coronary bioresorbable scaffold (BRS, Abbott Vascular, Santa Clara, USA) between May 2012 and May 2014 in our center were enrolled. Clinical/procedural characteristics and outcome data at 1868 (1641–2024) days were collected. The incidence of scaffold thrombosis (ScT), restenosis (ScR), and target lesion failure (TLF) and their predictors were investigated using Kaplan–Meier and Cox regression analysis. Results: 512 consecutive patients and 598 lesions were included in the database. A total of 30 ScT, 42 ScR, and 92 TLF were reported. The rate of ScT was 3.6% in the first year, 2.2% in the second–third year, and 0.6% in the fourth–fifth year after implantation. The corresponding rates of ScR were 2.5%, 5.7%, and 1.1%. The corresponding incidence of TLF was 8.8%, 8.0%, 3.8%. Procedural parameters (vessel size, scaffold footprint) and the technique used at implantation (including predilation, parameters of sizing, and postdilation) were predictors of ScT and TLF in the first three years after implantation. In contrast, only diabetes was predictive of events between 4–5 years (HR 6.21(1.99–19.40), *p* = 0.002). Conclusions: After device resorption, the incidence of very late adverse events in lesions/patients implanted with a BRS decreases. Procedural and device-related parameters are not predictors of events anymore.

## 1. Introduction

The Absorb bioresorbable coronary vascular scaffold (BRS) was the first of a new class of devices for the treatment of coronary artery disease. This poly-(L)-lactide acid-based scaffold with a coating of everolimus was designed to resorb completely ≈3 years after implantation, thus limiting the complications associated with a permanent foreign body, including vascular inflammation and the loss of physiologic motion of the vessel.

However, after the early results showing non-inferiority compared to latest generation everolimus-eluting stents, subsequent evidence of a significantly increased risk of scaffold thrombosis (ScT) and scaffold restenosis (ScR) was reported in all-comer registries and randomized control trials [1,2,3,4].

The mechanisms and predictors of these events have been thoroughly studied [5,6,7,8,9,10,11,12]. In our previous papers [7,8], we reported that, although a difference between periods was observed, the major predictors and mechanisms of events up to 3 years after BRS implantation were device and procedural parameters. For instance, studies [8,10] identified a small (<2.5 mm) reference vessel size as the principal mechanism of early (<30 day) ScT; in contrast, implantation in large (>3.5 mm) vessels was associated with late events, suggesting that malapposition and scaffold strut discontinuities during resorption might predispose to this event. We and others previously described how the mechanical/physical characteristics of first-generation scaffolds may explain these associations [6,7]. In these papers, consistent evidence was produced that the thicker struts of BRS might cause flow turbulence leading, particularly in settings of incompletely expanded scaffolds (early device thrombosis) or malapposition (late thrombosis), to both platelet aggregation and neointima proliferation and to an increased risk of events. Consistent with this, a number of studies reported that an optimal implantation technique including pre- and postdilation to optimize expansion as well as correct lesion and scaffold selection markedly reduced this risk [13,14].

Data from 5-years follow-up, including a meta-analysis of four randomized trials, are now available [15,16]. These data show that, after three years (i.e., after resorption), the excess of risk associated with BRS implantation as compared to drug eluting stent implantation disappears. The investigation of the incidence and mechanisms of target lesion events after resorption is, however, a key question, important for all devices in this class, that still needs to be addressed. The present study was designed to compare the incidence of events before resorption (divided in early (first year) and late (second to third year)) to that after resorption (fourth and fifth year after implant), to investigate the role of clinical and procedural predictors in each of these intervals, and ultimately to test whether device-, procedure-, lesion-, or patient-related characteristics influence the outcomes after device resorption.

## 2. Materials and Methods

### 2.1. Study Design

Patients who underwent percutaneous coronary intervention with at least one BRS (Absorb, Abbott Vascular, Santa Clara, CA, USA) from May 2012 to December 2014 in the catheterization laboratory of the University Hospital Mainz were consecutively enrolled in this all-comer registry. Patients who were eligible for 5 years follow-up were included in this analysis. The study is part of the Mainz IntracoronAry registry MICAT project (NCT02180178), which is approved by the ethics committee of Rheinland-Pfalz.

### 2.2. BRS Implantation

Details on inclusion criteria and selection of patients/lesion have already been published elsewhere [10]. Briefly, no BRS was implanted in lesions of the left main stem, in degenerated saphenous vein grafts, in-stent restenosis, in vessels visually estimated smaller than 2.25 mm or larger than 4 mm, and in true bifurcation lesions requiring two-stent techniques. Patients on chronic therapy with anticoagulants, allergy, or intolerance to aspirin or any other P2Y12 receptor antagonist and those with a limited life expectancy were excluded.

Heparin was administered peri-procedurally, the use of GP IIb/IIIa inhibitors or debulking devices were left to decide by the treating interventionalist.

Predilation was used in all cases, the systematic use of a PSP technique, however, was introduced as a local policy starting January 2014. After implantation, dual antiplatelet therapy for 12 months was prescribed for all patients comprising aspirin, clopidogrel, or ticagrelor/prasugrel for unstable settings, e.g., acute coronary syndrome.

### 2.3. Quantitative Coronary Analysis

Quantitative coronary analysis (QCA) was performed with Xcelera, R 4.1 (Philips, the Netherlands). Key measurements (before and after implantation) included the interpolated reference vessel diameter (RVD) and the (in-BRS) minimum lumen diameter (MLD). Residual stenosis was calculated by subtracting the quotient of the MLD and RVD from 1. Scaled stenosis was calculated by subtracting the quotient of the MLD and nominal scaffold size from 1.

The maximum footprint, expressing the maximum percentage of the circumference of the vessel occupied by struts, was calculated as:
$$\mathrm{footprint}=\frac{100*\mathrm{BRS}\;\mathrm{outer}\;\mathrm{surface}}{\mathrm{BRS}\;\mathrm{length}\;*\;3.14\;*\mathrm{MLD}\;}.$$

Lesions were classified according to the American Heart Association classification [17].

### 2.4. Endpoints

Follow-up was performed by trained personnel by computer-assisted telephone interview as previously published [10]. Events were adjudicated after review of original clinical data by two interventional cardiologists according to the academic research consortium definitions [18]. Target lesion failure (TLF) was defined as any cardiac death, target-vessel myocardial infarction, or clinically-driven target lesion revascularization.

Target lesion revascularization (TLR) was defined as any need of revascularization in the segment originally treated with a BRS. Scaffold thrombosis (ScT) was classified in early, late, and very late based on the timing of occurrence in case of definite or probable ScT.

### 2.5. Statistical Methods

Continuous data are described as mean and standard deviation or median and interquartile range and were compared using a parametric or nonparametric test based on the inspection of the Q–Q plots. Categorical data are described as total numbers and proportions and were analyzed with the Fisher´s exact-test.

Uni- and multivariate Cox regression analysis was used to describe the association between clinical/procedural parameters and outcome events.

Survival curves are presented as Kaplan–Meier curves with corresponding Cox P values.

Separate analyses were performed for the following events which occurred within 1 year after implantation, between two and three years, and between four and five years after implantation: scaffold thrombosis, scaffold restenosis, and target lesion failure. The impact of the following variables on outcomes during the first year, 2–3 years, and 4–5 years after implantation was assessed in univariable Cox regression analysis: cardiovascular risk factors including age, gender, hypertension, diabetes, smoking, dyslipidemia, renal function, history of prior revascularization via percutaneous coronary intervention (PCI) or coronary by-pass (CABG), prior stroke/transient ischemic attack (TIA), and left ventricular ejection fraction (LVEF), clinical presentation during index procedure, treated vessel, and location of lesion, optimal implantation technique and its components, lesion characteristics, such as RVD, MLD, sizing, scaled, and residual stenosis, MLD, length of scaffold, used antiplatelet medication, follow-up days. The optimal implantation technique was defined as previously published (briefly: vessel comprised between 2.5 and 3.5 mm, pre- and postdilation with balloon of at least the same size as the intended scaffold, postdilation at >12 atmospheres, no implantation in bifurcation or ostial lesions, stent to scaffold ratio comprised between 0.9 and 1.1, residual stenosis and residual scaled stenosis <20%) [19]. Variables associated with a *p* < 0.05 in univariate analysis were entered in multivariate analysis.

All analyses should be considered exploratory. Data were analyzed with MedCalc (Version 15.8, Ostend, Belgium).

## 3. Results

### 3.1. Patient Characteristics

A total of 512 patients with 598 lesions of the MICAT registry were eligible for 5 years follow-up on 1 May 2019. The characteristics of these patients are presented in Appendix A. Median age was 62 (54–73) years, 78.7% of the patients were male, 70.7% had hypertension or was on antihypertensive medication, 37.1% were dyslipidemic and/or were on medication treatment with statins, 42.6% were smokers, and 19.9% suffered from diabetes. Patients with a history of PCI were 26.4% of the total, those with prior stroke or TIA were 3.3%. Median estimated glomerular filtration rate (eGFR) was 83 mL/min/1.73m² (69–99.5) and median LVEF was 55% (50–55%). With regards to the clinical presentation, 12.1% of the patients presented with unstable angina, 29.5% with non-ST elevation myocardial infarction (NSTEMI), and 25.4% as STEMI; 32.4% presented with stable or silent angina.

### 3.2. Lesion Characteristics

The target vessel was the left anterior descending (LAD) artery in 44.8%, the right coronary artery (RCA) and left circumflex artery (LCX) in 28.9% and 26.1% of the cases, respectively.

Ostial and bifurcation lesions were revascularized in 8.7% and 13.2% of the cases, respectively. The prevalence of chronic total occlusions (CTO) was 2.8%, 41.3% of the lesions were a complex B2 or C type lesion.

The median total stented length per patient was 18 mm (18–30 mm). The mean number of vessels treated with scaffolds per patient was 1.2 ± 0.5, the mean number of scaffolds implanted per patient was 1.4 ± 0.9. The mean of total stented length per lesion was 24.1 ± 13.4 mm.

### 3.3. Lesion Treatment and Immediate Angiographic Results

Appendix A shows lesion and angiographic results. Predilation was performed almost systematically (98.3%). The minimum inflation pressure of scaffold deployment per lesion was 13.6 ± 1.9 ATM.

Postdilation was performed in 35.1% of the lesions with 15.1 ± 3.7 ATM.

The ratio of the minimal lumen diameter after implantation to the nominal BRS diameter, expressing BRS deployment, was 0.8 ± 0.2. Maximum footprint was 37% (34–43).

Among the lesions treated, in 11.5% of the patients, a BRS was implanted overlapping with a close-by stent or scaffold.

An “optimal implantation technique” was used in 214 lesions of 205 patients (35.8% of all patients, 40.0% of all lesions).

### 3.4. Follow-Up

The median follow-up was 1868 (1641–2024) days. A lesion-oriented 5-years follow-up was available in 410 of 512 (80%) eligible patients. Table 1 shows the number of events and the Kaplan–Meier estimates of the observed endpoints of scaffold thrombosis (ScT), clinical scaffold restenosis (ScR), and target lesion failure (TLF). In total, 30 definite or probable ScT occurred during follow-up, of which 13 were acute or subacute and 17 were late or very late thrombosis. The corresponding Kaplan–Meier estimates for ScT were 3.6% in the first year, and 2.2% in the interval 2–3 years, and 0.6% in the fourth–fifth year. In total, there were 42 patients who suffered from scaffold restenosis of which 12 occurred in the first year, 26 between 2–3 years, and 4 between 4 and 5 years of follow-up, respectively (yearly KM rates 2.5%, 4.3%, 1.4%, 1.1%, and 0%, respectively).

A total of 92 patients reached the endpoint TLF, in the first year 44, between 2–3 years, 35 patients (yearly KM rates 8.8% in the first year, 6.1% in the second, and 2.0% in the third).

In the time interval of 4–5 years, the endpoint TLF was reached by 13 patients (yearly KM rate 1.7% in fourth and 2.1% in the fifth year), which was mainly driven by the occurrence of target vessel-myocardial infarction (TV-MI). There were 2 cardiovascular deaths, 3 ischemia-driven TLRs, and 8 TV-MIs.

Figure 1a–c depicts the Kaplan–Meier survival curves of ScT, ScR, and TLF during the three time intervals.

### 3.5. Predictors of Target Lesion Failure

Table 2 shows the associations of the pre-specified predictors of TLF in univariate Cox regression analysis for the time intervals of one year, 2–3 years, and 4–5 years, respectively.

The technique used at the time of implantation was a predictor of TLF both in the first and in the interval 2–3 years after index (HR 0.44 (0.22–0.88), *p* = 0.02, and HR 0.34 (0.17–0.66), *p* = 0.01).

In contrast, the use of an optimal implantation technique was not associated with the incidence of events during 4–5 years after BRS implantation (HR 0.91 (0.30–2.74), *p* = 0.86).

In the first year after implantation, the maximum footprint and an RVD < 2.5mm were predictors for the incidence of TLF (HR 1.06 (1.04–1.08), *p* < 0.0001, HR 1.99 (1.03–3.86), *p* = 0.04, respectively).

An RVD smaller than 2.5 mm and an RVD larger than 3.5 mm were predictors for TLF in the time period of 2–3 years (HR 2.17 (1.05–4.50) and HR 2.33 (1.06–5.12), both *p* = 0.04), along with footprint (HR 1.09 (1.06–1.11), *p* < 0.0001).

None of the clinical characteristics, except for diabetes, was a predictor of TLF in the interval after 2–3 years post implantation (HR 2.26 (1.13–4.53), *p* = 0.02) (Appendix A).

Figure 2 depicts the Kaplan–Meier survival curves of TLF in each of the time intervals divided by patients who received scaffold implantation in a vessel with a RVD smaller 2.5 mm and larger than 3.5 mm, (upper and lower panels, respectively).

Interestingly, the incidence of TLF was higher in small vessels up to the third year, but not thereafter (Figure 2, upper panel). Moreover, a reference vessel diameter >3.5 mm was a risk factor for target lesion failure during the second and third year only (Figure 2, lower panel).

As opposed to the two earlier time intervals, the optimal implantation technique could not be shown as a predictor for very late TLF 4–5 years after implantation (HR 0.91 (0.30–2.74), *p* = 0.86). Also, an RVD smaller 2.5 mm or larger 3.5 mm were not predictors of TLF in the time of 4–5 years. (HR 1.03 (0.23–4.60), *p* = 0.97, HR 0.64 (0.08–4.89), *p* = 0.67, respectively). Figure 2 presents the corresponding overall KM curves.

In univariate Cox regression analysis for the period 4–5 years, including all variables listed in Appendix A, only the classic cardiovascular risk factors diabetes (HR 7.50 (2.47–22.82), *p* = 0.0004) and impaired renal function (HR 0.97 (0.95–0.99), *p* = 0.02) were predictors for the incidence of TLF (Appendix A).

In multivariate Cox regression model incorporating both eGFR and diabetes, the only independent predictor associated with TLF during the 4–5 years interval was diabetes (HR 6.21 (1.99–19.40), *p* = 0.002) (Appendix A).

## 4. Discussion

We report on the long-term outcomes after BRS implantation in an all-comer cohort of patients with coronary artery disease.

The main findings of the present study can be summarized as follows:-the incidence of late adverse events after implantation of a BRS decreases over the time.-the procedural and angiographic characteristics that could be identified as predictors of negative outcome in the first three years after index procedure do not predict TLF between 4–5 years, and diabetes, a known patient-related risk factor, was the only predictor of TLF between 4–5 years.

### 4.1. Target Lesion Failure in Stent-Studies

Bioresorbable scaffolds were introduced based on the concept that a permanent implant may represent a continuous stimulus for inflammation, platelet activation, and neointima proliferation. In previous publications, the mechanisms and predictors of events up to ≈3 years after BRS implantation were described [5,6,7,8,9,10,11,12]. We extend this information with an analysis of the incidence and predictors of events up to 5 years in the present paper. To our knowledge, this is the first paper investigating the existence of differences in the predictors of events among the different time periods. The decrease in the event rates in the 4–5 years interval appears to support the validity of the concept of resorbable devices. Although the single-arm nature of this study does not allow comparisons, previous stent studies reported incidence rates similar to those described here. Pilgrim et al. reported a TLF rate of approximately 2.7% in the fifth year and 3.5% after the fourth in the BIOSCIENCE trial investigating an ultra-thin strut, biodegradable polymer sirolimus-eluting stent and a thin-strut, durable polymer everolimus-eluting stent [19]. Further, in the 10 years outcome data of the ISAR-Test 4 trial, comparing a biodegradable polymer-based sirolimus eluting stent with a permanent polymer everolimus-eluting stent, a TLF incidence of approximately 2.5% after four and five years was observed [20]. Although an indirect comparison is impossible, these rates are nearly 50% higher than those observed in our study.

Similar TLF rates in DES and BRS were also shown in a meta-analysis comparing three to four years outcome data from 4 randomized trials (*n* = 3245, BVS = 2075, DES = 1170). In this analysis, no significant difference with respect to TLF or any other of the adverse events was found [21].

In sum, the very low rate of target lesion failure appears to support the concept that the absence of a permanent implant might be associated with a reduce rate of events late after bioresorption of the device.

### 4.2. Predictors of Early and Late Events

A number of previous publications, including those of our group, reported that device and procedural parameters including vessel size, use of pre- and postdilation, device to reference vessel diameter ratio, are major predictors of events up to three years [5,6,7,8,9,10,11,12,22,23]. The current data suggest that only diabetes, a well-known risk factor for late events after stenting [24], is a determinant of very long-term outcome, and that the incidence of outcomes is independent of device/procedural characteristics. This evidence has implications for the concept of BRS: for instance, both under- and oversizing are associated with increased neointima proliferation. The evidence that these parameters do not predict 4–5 years events suggests that these phenomena are not relevant upon long term. Further, the fact that vessel characteristics (e.g., small vessel size) did not predict events after BRS resorption suggests that the endothelialization and neointima formation induced by BRS (despite the thick struts) do not compromise blood flow upon long term.

### 4.3. Limitations

Even though the methods used for the acquisition and analysis of the data were similar to those used in randomized controlled trials, the results of this single-center registry need to be taken as exploratory. However, given the absence of in- and exclusion criteria beyond the device´s instructions for use, the current data provide an accurate reflection of daily routine. Also reflecting daily routine, the rate of lost to follow-up was unfortunately relatively high in the current cohort.

## 5. Significance and Future Perspectives

We present data on the very long-term outcome of patients treated with BRS. Since BRS are expected to be fully resorbed at this time point, our data are likely to be valid for all current (and future) resorbable devices. In the 4–5 years interval, the incidence of target lesion failure and correlated adverse events in lesions/patients treated with a BRS dropped substantially, suggesting that vessel healing has occurred at this time. Further, while procedural and device-related parameters were the major predictors of events in the first year and in the interval 2–3 years, diabetes was the only predictor in the fourth and fifth year. This observation supports the concept that events during the 4–5 years follow-up are independent of the device used to treat the index stenosis and are simply related to the risk profile of the patient. Nonetheless, the cumulative 5-year incidence of scaffold thrombosis remained unacceptably high. Collectively, these data appear to validate the concept of BRS, and emphasize the need of second-generation devices with a better profile of early safety.

## Figures and Tables

**Figure 1 jcm-09-00847-f001:**
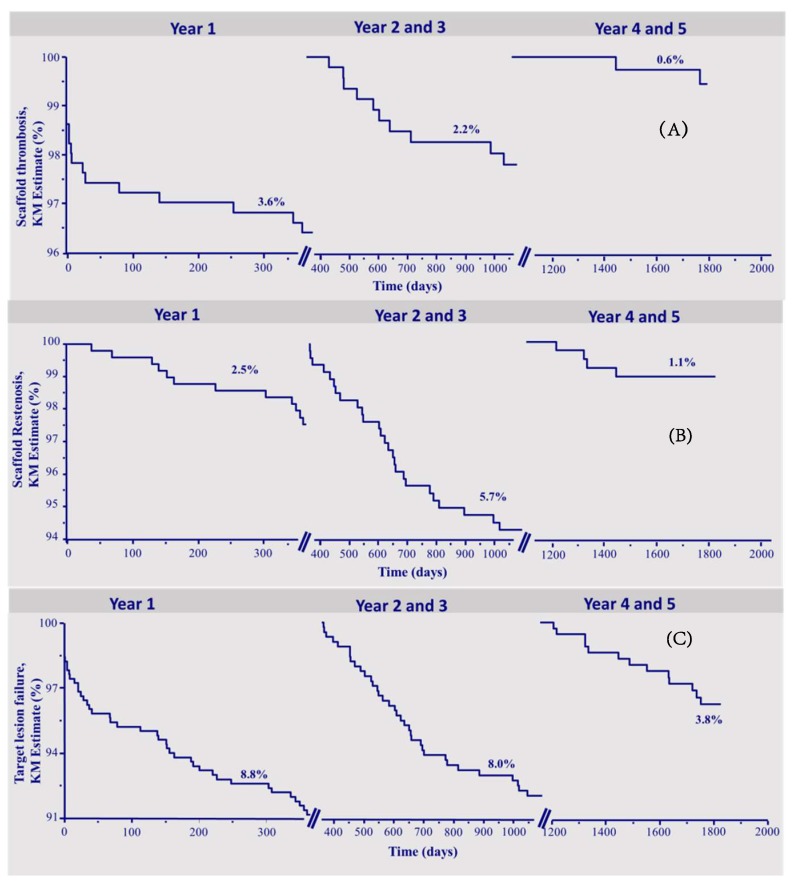
Kapla–Meier survival curves. (**A**) Scaffold thrombosis. (**B**) Scaffold restenosis. (**C**) Target lesion failure.

**Figure 2 jcm-09-00847-f002:**
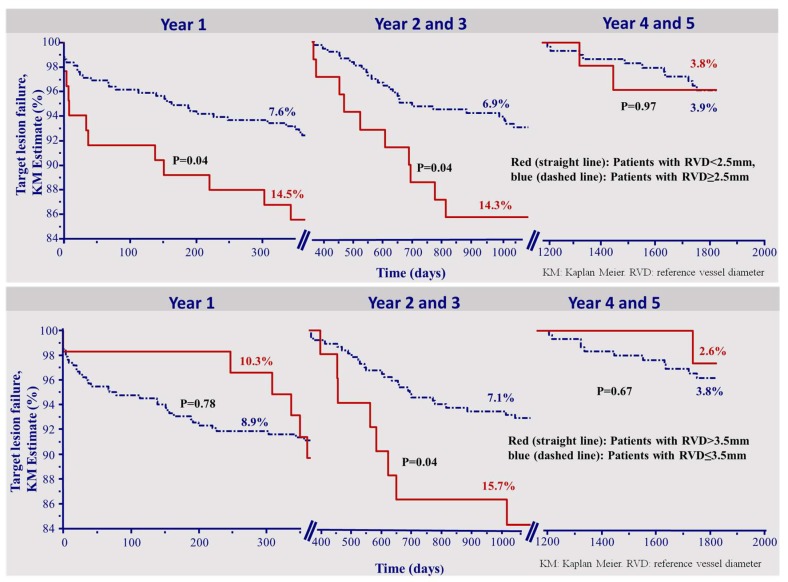
Kapla–Meier survival curve for the incidence of target lesion failure. The incidence of target lesion failure was higher in small vessels (RVD < 2.5 mm) up to 3 years, but not thereafter (upper panel). Finally, the incidence of TLF was initially (<1 year) and then higher (2–3 years) in vessels larger than 3.5 mm. After 3 years, this parameter had no impact (lower panel). KM: Kaplan–Meier. RVD: reference vessel diameter.

**Table 1 jcm-09-00847-t001:** Number of events and annualized Kaplan-Meier (KM) risk of adverse events divided by patients with and without optimal implantation and respective hazard ratios (HR) in univariate Cox regression analysis during whole observation period of 5 years. TLF: target lesion failure; ScR: scaffold restenosis; ScT; scaffold thrombosis. **Bold** character identifies parameters associated with *p* values < 0.05.

**Within 1 Year**	**All Patients**	**Optimal Implantation**	**Non-Optimal Implantation**	***p***	**HR** **95% CI**
ScT	18 (3.6%)	2 (1%)	16 (5.3%)	0.02	**5.36** **(1.31–8.65)**
ScR	12 (2.5%)	2 (1.0%)	10 (3.4%)	0.10	3.29 (0.82–8.31)
TLF	44 (8.8%)	10 (5.1%)	34 (11.3%)	0.02	**2.29** **(1.14–3.80)**
**2–3 years**	**All Patients**	**Optimal Implantation**	**Non-Optimal Implantation**	***p***	**HR** **95% CI**
ScT	10 (2.2%)	2 (1.1%)	8 (2.9%)	0.18	2.78 (0.67–8.39)
ScR	26 (5.7%)	5 (3.7%)	21 (7.7%)	0.03	**2.90** **(1.12–5.34)**
TLF	35 (8.0%)	7 (3.9%)	28 (10.8%)	0.01	**2.95** **(1.28–4.91)**
**4–5 years**	**All Patients**	**Optimal Implantation**	**Non-Optimal Implantation**	***p***	**HR** **95% CI**
ScT	2 (0.6%)	2 (1.4%)	0	0.08	-
ScR	4 (1.1%)	2 (1.3%)	2 (0.9%)	0.67	0.65 (0.09–4.79)
TLF	13 (3.8%)	5 (3.5%)	8 (4.0%)	0.86	1.10 (0.36–3.33)
**Cumulative** **5 years**	**All Patients**	**Optimal Implantation**	**Non-Optimal Implantation**	***p***	**HR** **95% CI**
ScT	30 (5.9%)	6 (3.5%)	24 (8.1%)	0.020	**2.76** **(1.33–5.72)**
ScR	42 (8.2%)	9 (5.0%)	33 (11.6%)	0.010	**2.54** **(1.37–4.70)**
TLF	92 (19.3%)	22 (12%)	70 (24.0%)	0.0005	**2.30** **(1.52–3.48)**

**Table 2 jcm-09-00847-t002:** Predictors of TLF in univariate Cox regression analysis during different time intervals of follow-up duration. HR: Hazard ratio. RVD: reference vessel diameter. **Bold** character identifies parameters associated with *p* values < 0.05.

Max Footprint	HR	*p*
**1 year**	**1.06 (1.04–1.08)**	**<0.0001**
**2–3 years**	**1.09 (1.06–1.11)**	**<0.0001**
4–5 years	1.03 (0.96–1.11)	0.40
Optimal Implantation		
**1year**	**0.44 (0.22–0.88)**	**0.02**
**2–3 years**	**0.34 (0.17–0.66)**	**<0.01** **(0.007)**
4–5 years	0.91 (0.30–2.74)	0.86
RVD < 2.5 mm		
**1 year**	**1.99 (1.03–3.86)**	**0.04**
**2–3 years**	**2.17 (1.05–4.50)**	**0.04**
4–5 years	1.03 (0.23–4.60)	0.97
RVD > 3.5 mm		
1 year	1.13 (0.48–2.67)	0.78
**2–3 years**	**2.33 (1.06–5.12)**	**0.04**
4–5 years	0.64 (0.08–4.89)	0.67

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
