# Peer review of "Five Years Outcomes and Predictors of Events in a Single-Center Cohort of Patients Treated with Bioresorbable Coronary Vascular Scaffolds"

_jcm, 2020, doi:10.3390/jcm9030847_

Round 1

Reviewer 1 Report

I am happy with the corrections

Author Response

 Reviewer´s comment

I am happy with the corrections.

Authors’ Answer: We thank the Reviewer for taking the time to evaluate our manuscript. Thank you for the positive feedback.

Reviewer 2 Report

A review with great pleasure this work by Anadol et al. Authors report outcome data of patients treated with coronary bioresorbable scaffolds up to 5 14 years and investigate predictors of adverse events. Several methodological limitations need to be addressed.

Abstract: Please double check that the current abstract meets guidelines for the journal. Also there seems to be a difference in font in the conclusion.

Introduction: fails to provide the reader with background on the importance of CAD CVD   and coronary vascular scaffolds, where is this useful why is it you why is it need it. The current version of the introduction fails to provide sufficient background. The manufacture of the Scaffolds do not need to to be presented in the introduction.

Study design: fails to provide sufficient details. Did the patient complete consent? Were there any difference between those that did not sign consent ? Where there any

Differences in those that were excluded because of lack of follow up. How was follow up assessed and how was completeness assessed?

How was missing data handled?

Statistical analysis:

Kaplan Meier analysis to be presented before cox regression models. 

Please clarify what separate outcome analysis means in the current version of the manuscript.

Is this KM event rates at pre specified time points? Or does this represent censoring?

Please clarify how did adjustment variables from the Cox models were selected.

The previous comments need to be Address before the rest of the information provided in the manuscript can be interpreted.

Thank you for allowing me to review this work.

Author Response

Reviewer #2

A review with great pleasure this work by Anadol et al. Authors report outcome data of patients treated with coronary bioresorbable scaffolds up to 5 14 years and investigate predictors of adverse events. Several methodological limitations need to be addressed.

Authors’ Answer: We thank the Reviewer for taking the time to evaluate our manuscript. We have tried to address them all in a point-by-point list, which you will find below.

Abstract: Please double check that the current abstract meets guidelines for the journal. Also there seems to be a difference in font in the conclusion.

Authors’ Answer: We thank the Reviewer for this comment. The format of the abstract complies with the instructions to authors (“A single paragraph of about 200 words maximum. …. We strongly encourage authors to use the following style of structured abstracts, but without headings”). However, we reformatted it to make it more readable. As suggested, we used the same font in the whole abstract.

Introduction: fails to provide the reader with background on the importance of CAD CVD and coronary vascular scaffolds, where is this useful why is it you why is it need it. The current version of the introduction fails to provide sufficient background. The manufacture of the Scaffolds do not need to be presented in the introduction.

Authors’ Answer: We thank the Reviewer for this comment. As suggested, we modified the introduction section to better describe the background and goal of this study. We also removed the sentence about the manufacture of the BRS according to reviewer’s comment.

Study design: fails to provide sufficient details. Did the patient complete consent? Were there any difference between those that did not sign consent?

Authors’ Answer: We thank the Reviewer for this comment. Details on inclusion criteria and selection of patients/lesion have been published previously (reference 10). All the patients included in the study signed the consent.

Where there any differences in those that were excluded because of lack of follow up. How was follow up assessed and how was completeness assessed? How was missing data handled?

Authors’ Answer: We thank the Reviewer for this comment. Follow-up was performed by trained personnel by computer-assisted telephone interview as previously published (reference 10: if the patient was not found, at least 5 attempts were performed. The treating family physician or cardiologist was also contacted. In case of patients who were admitted to other hospitals, original documents were collected unless it was clear that the cause of admission was non-cardiac. All events were adjudicated based on original documents by consensus of two expert cardiologists). The median follow-up was 1868 (1641-2024) days. A lesion-oriented 5-years follow-up was available in 410 of 512 (80%) eligible patients. This is reported in the results. We added the lack of completeness of follow-up as a limitation of this study as follows: “Also reflecting daily routine, the lost to follow-up rate was unfortunately relatively high in the current cohort”.

Statistical analysis:

Kaplan Meier analysis to be presented before cox regression models.

Authors’ Answer: This has been done. The exception is Figure 2, which presents a Kaplan Meier analysis to graphically depict the results of the Cox analysis.

Please clarify what separate outcome analysis means in the current version of the manuscript.

Authors’ Answer: We simply meant to state that separate analyses were performed (e.g., an analysis for ScT, one for TLF, etc). Sorry for the misunderstanding.

Is this KM event rates at pre specified time points? Or does this represent censoring?

Authors’ Answer: These were pre-specified endpoints.

Please clarify how did adjustment variables from the Cox models were selected.

Authors’ Answer: We thank the Reviewer for this comment. We now added this sentence to the methods: Variables associated with a P<0.05 in univariate analysis were entered in multivariate analysis.

We also tested the effect of entering age (the only variable with P<0.1), but this did not change the results.

The previous comments need to be Address before the rest of the information provided in the manuscript can be interpreted.

Reviewer 3 Report

This article belongs to a series of articles published by the authors on single-center cohort patients treated with bioresorbable coronary vascular scaffolds. In my opinion, the article does not provide enough interesting data, that cannot be obtained from other published research articles and/or metanalsysis.

General comments

1) Overall, it appears as if this manuscript was put together from bits and pieces from here and there. The authors have not devoted enough attention to details, making it difficult to follow its logic. In addition, the manuscript requires extensive work, thorough typos editing, format and removal of inconsistencies between data in the text with the tables and figures.

2) Format and font must be the same throughout the manuscript

3) Interval vs. year description, the authors should decide how to present the data. It is confusing that data is presented in year intervals in some results, and by year in another.

4) The data should be properly mentioned in both the manuscript and the corresponding figures and tables, and the other way around.

5) The article should focused on the importance of the data rather than merely a description of outcome data.

Major comments

The abstract needs to be rewritten in order to facilitate flow to the reader (introduction, methods, aims, results and general conclusion) Please, include SCR data in the abstract. The data presented in the abstract does not match the KM of the table. Some results are presented by year, however in the table the data is presented by intervals. This is confusing. Introduction: it is not clear from the introduction what is the hypothesis, or why this work was performed. Line 133-135: The data on mean number of vessels treared with scaffold per patient and the mean number of scaffolds implanted per patient does in the text differ from the contents in supplementary table 2. Please, explain. For consistency: if in the table 1 the data is presented as TLF, ScR and ScT, the corresponding explanatory text should follow the same order, however, in the text the authors first mentioned ScT, ScR and TLF. In the abstract the authors stated that outcome data was collected at 1868 (1641-2024), whereas in section 3.4, the authors stated that follow up was 1868 (1640.5-2024). For consistency, please round up the numbers accordingly. For consistency, decide if the data is presented by year or by intervals both in the table, manuscript and abstract.Line 153. Data in the text is presented per year, however in the table is presented in intervals (1, 2-3, 4-5, and 5). This is confusing for the reader. The corresponding Kaplan-Meier estimates for ScT were 3.6% in the first year, and 1.7%, 0.5%, 0% and 0.6% in the second, third, fourth and fifth year. However, the data in the table says otherwise, the estimates for ScT are 3.6% in the first year, and 2.2% for second and third, and 0.6% for 4 and 5 year, and 5.9% the 5th year. Line 156. Same as above. Please, for consistency. The yearly rates presented in the text: yearly KM rates 2.5%, 4.3%, 1.4%, 1.1% and 0%, respectively. Cannot be found in table 1 because the table describe time intervals. Please decide. Line 161. Same as above. Figure 1. The authors included a Kaplan meier survival curves for the whole 5-year follow-up period. However, I find again inconsistencies. Figure 1, ScR in figure 1A is 9%, in table 1 is 8.2%. Figure 1B, ScT in figure 1B is 5.3% and in the table is 5.9%. In table 1C, TLF is 23%, and in the table is 19.3%. Line 179. The authors stated that the predictor of TLF for the interval of 2-3 years is HR 0.34 [0.15-0.77], p=0.01. However, in table 2, the HR ratio presented is HR 0.34 [0.17-0.66], p<0.01. Table 2 does not show data on ScR and Sct despite the fact that the first sentence of the section states it shows the associations in table 2. Again, the authors indicate that optimal implantation technique was not associated with the incidence of events during 4-5 years after BRS implantation HR 0.91 [0.30-2.76] p=0.86. However, in the table, the data is HR 0.91 [0.30-2.74] p=0.86. Figure 2A is redundant as the data is already presented in table 1. Please remove. In figure 2B, the figure indicates that red and blue line are patients with RVD > or < than 2.5 mm. However, in the text the authors write a vessel diameter of > or < than 3.5 mm.The text also indicates that the data presented is for 2 and 3 year only. However, in the figure the authors include from year 1 to 4 and 5. Please, clarify. Since the study does not allow comparisons, the authors should re-write their findings as follows:The incidence of late adverse events after implantation of a BRS decreases over the time.The rest of the paragraph should be moved to the corresponding section in the discussion:-  ‘values similar to those expected in patients treated with drug eluting stents’.- The authors should mention that the following text is an hypothesis, and not part of the main findings, and cite accordingly: the decrease in the rate of lesion re-stenosis might be compatible with the concept that the absence of a foreign body might reduce the trigger for neointima/media proliferation. The discussion section needs to be rewritten in order to facilitate flow, and not just data description. Furthermore, the authors should emphasize the difference between this work and the previous work by the authors: Polimeni, A.; Weissner, M.; Schochlow, K.; Ullrich, H.; Indolfi, C.; Dijkstra, J.; Anadol, R.; Münzel, T.; Gori, T. Incidence, Clinical Presentation, and Predictors of Clinical Restenosis in Coronary Bioresorbable Scaffolds. JACC Cardiovasc. Interv. 2017, 10 (18), 1819–1827. Anadol, R.; Schnitzler, K.; Lorenz, L.; Weissner, M.; Ullrich, H.; Polimeni, A.; Münzel, T.; Gori, T. Three-Years Outcomes of Diabetic Patients Treated with Coronary Bioresorbable Scaffolds. BMC Cardiovasc. Disord.2018, 18 (1), 92.

Minor comments

Line 15: there is no need to include patient description here. Line 17: there is no need to include the month as presented 5-2012 and 5-2014, for that the M&M is for. Line 40: please replace studied. [5-12] by studied [5-12]. Line 44-47: please clarify the association described in the following paragraph since the paragraph is not clear for the reader. Compatible with the importance of procedural parameters, a number of studies reported an association between the implementation of an optimal implantation technique, comprising of pre- and postdilation as well as correct lesion and scaffold selection (so-called PSP) and a reduced risk of adverse events.[13, 14] Line 93: format accordingly in color black Line 99: please include full name and abbreviations - Propensity score (PS). Line 118-126: Please, re-write the paragraph, some sentences are well elaborated others are short and simple. Do not start the sentence with a percentage value. Line 133-135: Some data from the paragraph appears extracted from supplemental table 2. Please cite accordingly. Line 167: format accordingly in color black Line 206: the authors should include the diabetes results in the table.

Author Response

Reviewer #3.

This article belongs to a series of articles published by the authors on single-center cohort patients treated with bioresorbable coronary vascular scaffolds. In my opinion, the article does not provide enough interesting data, that cannot be obtained from other published research articles and/or metanalsysis.

Authors’ Answer: We thank the Reviewer for taking the time to evaluate our manuscript. We have now quoted 2 previous publications with 5-years data. In our opinion, the manuscript does indeed provide new long-term data that could be of interest, particularly with regards to the predictors of adverse events during the different observation intervals. To our knowledge, none of the previous 5-years follow-up papers has provided a differential analysis of the predictors of events. The results of this analysis, which we discuss in paragraph 5, are indeed important because they suggest that the mechanisms of the (relatively rarer) events during the 4-5 years follow-up are not related to the device implanted or the lesion treated, but to the general risk profile of the patient. This information is critical for the concept of BRS (and eventually for future devices of this class).

General comments

1) Overall, it appears as if this manuscript was put together from bits and pieces from here and there. The authors have not devoted enough attention to details, making it difficult to follow its logic. In addition, the manuscript requires extensive work, thorough typos editing, format and removal of inconsistencies between data in the text with the tables and figures.

Authors’ Answer: We thank the Reviewer for this comment. As suggested, we did an extensive revision of typos, format and inconsistencies of the paper. We will be happy to correct any additional error the reviewer identifies. 

2) Format and font must be the same throughout the manuscript

Authors’ Answer: We thank the Reviewer for this comment. As suggested, we did an extensive revision of the format and we use the same font in the whole manuscript. The font size varies according to the instructions to authors.

3) Interval vs. year description, the authors should decide how to present the data. It is confusing that data is presented in year intervals in some results, and by year in another.

Authors’ Answer: We thank the Reviewer for this comment. Incidence rates were presented in year intervals only in the text for full disclosure. This has now been deleted. As suggested, we presented the data only by intervals (1 year; 2-3 years: 4-5 years).

4) The data should be properly mentioned in both the manuscript and the corresponding figures and tables, and the other way around.

Authors’ Answer: We thank the Reviewer for this comment. As suggested, all data are presented in tables, figures and manuscript, as appropriate.

5) The article should focused on the importance of the data rather than merely a description of outcome data.

Authors’ Answer: We thank the Reviewer for this comment. As suggested, we discussed the importance and the impact of our data in a new paragraph entitled “Significance and future perspectives”.

Major comments

The abstract needs to be rewritten in order to facilitate flow to the reader (introduction, methods, aims, results and general conclusion) Please, include SCR data in the abstract.

Authors’ Answer: We thank the Reviewer for this comment. The format of the abstract complied with the instructions to authors. We rewrote it according reviewer’s suggestion and we included the data on Scaffold restenosis (SCR).

The data presented in the abstract does not match the KM of the table. Some results are presented by year, however in the table the data is presented by intervals. This is confusing.

Authors’ Answer: We thank the Reviewer for this comment. We apologize for these inconsistencies, which we have corrected in the revised manuscript and we presented the data only by intervals (1 year; 2-3 years: 4-5 years).

Introduction: it is not clear from the introduction what is the hypothesis, or why this work was performed.

Authors’ Answer: We thank the Reviewer for this comment. We now clearly state the aim of the present study in the abstract and in the introduction.

Line 133-135: The data on mean number of vessels treared with scaffold per patient and the mean number of scaffolds implanted per patient does in the text differ from the contents in supplementary table 2. Please, explain.

Authors’ Answer: We thank the Reviewer for this comment. These two endpoints were presented as median [IQR] in the table and mean[SD] in the text. This has been corrected in the revised manuscript.

For consistency: if in the table 1 the data is presented as TLF, ScR and ScT, the corresponding explanatory text should follow the same order, however, in the text the authors first mentioned ScT, ScR and TLF.

Authors’ Answer: The sequence has been corrected throughout the manuscript: ScT, ScR, TLF.

In the abstract the authors stated that outcome data was collected at 1868 (1641-2024), whereas in section 3.4, the authors stated that follow up was 1868 (1640.5-2024).

Authors’ Answer: Corrected.

For consistency, decide if the data is presented by year or by intervals both in the table, manuscript and abstract.

Authors’ Answer: As mentioned above, this has now been corrected.

Line 153. Data in the text is presented per year, however in the table is presented in intervals (1, 2-3, 4-5, and 5). This is confusing for the reader. 

Authors’ Answer: As mentioned above, this has now been corrected.

The corresponding Kaplan-Meier estimates for ScT were 3.6% in the first year, and 1.7%, 0.5%, 0% and 0.6% in the second, third, fourth and fifth year. However, the data in the table says otherwise, the estimates for ScT are 3.6% in the first year, and 2.2% for second and third, and 0.6% for 4 and 5 year, and 5.9% the 5thyear.

Authors’ Answer: This is correct. Since some of the patients were lost to follow-up during the 5 years, or reached the endpoint, it is indeed possible that the sum of the individual incidence rates does not match the overall (5 years) incidence rate.

Line 156. Same as above. Please, for consistency. The yearly rates presented in the text: yearly KM rates 2.5%, 4.3%, 1.4%, 1.1% and 0%, respectively. Cannot be found in table 1 because the table describe time intervals. Please decide. Line 161. Same as above.

Authors’ Answer: As mentioned above, this has now been corrected

Figure 1. The authors included a Kaplan meier survival curves for the whole 5-year follow-up period. However, I find again inconsistencies. Figure 1, ScR in figure 1A is 9%, in table 1 is 8.2%. Figure 1B, ScT in figure 1B is 5.3% and in the table is 5.9%. In table 1C, TLF is 23%, and in the table is 19.3%.

Authors’ Answer: All these are now presented as time intervals. We also introduced a much more informative figure divided in the three endpoints and the three periods of observation.

Line 179. The authors stated that the predictor of TLF for the interval of 2-3 years is HR 0.34 [0.15-0.77], p=0.01. However, in table 2, the HR ratio presented is HR 0.34 [0.17-0.66], p<0.01.

Authors’ Answer: Sorry, corrected.

Table 2 does not show data on ScR and Sct despite the fact that the first sentence of the section states it shows the associations in table 2.

Authors’ Answer: Paragraph 3.5 focuses on TLF. We had added the sentence on ScR and ScT as additional comment, but it has been now removed for clarity.

Again, the authors indicate that optimal implantation technique was not associated with the incidence of events during 4-5 years after BRS implantation HR 0.91 [0.30-2.76] p=0.86. However, in the table, the data is HR 0.91 [0.30-2.74] p=0.86.

Authors’ Answer: Sorry, this has been corrected.

Figure 2A is redundant as the data is already presented in table 1. Please remove. In figure 2B, the figure indicates that red and blue line are patients with RVD > or < than 2.5 mm. However, in the text the authors write a vessel diameter of > or < than 3.5 mm. The text also indicates that the data presented is for 2 and 3 year only. However, in the figure the authors include from year 1 to 4 and 5. Please, clarify.

Authors’ Answer: We thank the Reviewer for this comment. Figure 2A has been removed. We have modified the legend to make it more clear, the data and legend are correct.

Since the study does not allow comparisons, the authors should re-write their findings as follows: The incidence of late adverse events after implantation of a BRS decreases over the time. The rest of the paragraph should be moved to the corresponding section in the discussion:-  ‘values similar to those expected in patients treated with drug eluting stents’.- The authors should mention that the following text is an hypothesis, and not part of the main findings, and cite accordingly: the decrease in the rate of lesion re-stenosis might be compatible with the concept that the absence of a foreign body might reduce the trigger for neointima/media proliferation. The discussion section needs to be rewritten in order to facilitate flow, and not just data description. Furthermore, the authors should emphasize the difference between this work and the previous work by the authors: Polimeni, A.; Weissner, M.; Schochlow, K.; Ullrich, H.; Indolfi, C.; Dijkstra, J.; Anadol, R.; Münzel, T.; Gori, T. Incidence, Clinical Presentation, and Predictors of Clinical Restenosis in Coronary Bioresorbable Scaffolds. JACC Cardiovasc. Interv. 201710 (18), 1819–1827. Anadol, R.; Schnitzler, K.; Lorenz, L.; Weissner, M.; Ullrich, H.; Polimeni, A.; Münzel, T.; Gori, T. Three-Years Outcomes of Diabetic Patients Treated with Coronary Bioresorbable Scaffolds. BMC Cardiovasc. Disord.201818 (1), 92.

Authors’ Answer: We thank the Reviewer for this suggestion. We have emphasized the differences between this work and previous ones from our and other groups in the discussion (4.1 and 4.2).

Minor comments

Line 15: there is no need to include patient description here. Line 17: there is no need to include the month as presented 5-2012 and 5-2014, for that the M&M is for. Line 40: please replace studied. [5-12] by studied [5-12]. Line 44-47: please clarify the association described in the following paragraph since the paragraph is not clear for the reader. Compatible with the importance of procedural parameters, a number of studies reported an association between the implementation of an optimal implantation technique, comprising of pre- and postdilation as well as correct lesion and scaffold selection (so-called PSP) and a reduced risk of adverse events.[13, 14]

Authors’ Answer:  This section has been expanded, thank you.

Line 93: format accordingly in color black

Authors’ Answer:  All is in black.

Line 99: please include full name and abbreviations - Propensity score (PS).

Authors’ Answer: Thank you. P refers to statistical P, not propensity score.

Line 118-126: Please, re-write the paragraph, some sentences are well elaborated others are short and simple. Do not start the sentence with a percentage value. Line 133-135: Some data from the paragraph appears extracted from supplemental table 2. Please cite accordingly. Line 167: format accordingly in color black Line 206: the authors should include the diabetes results in the table.

Authors’ Answer: We thank the Reviewer for this comment. We revised the paragraph and the format according to reviewer’s suggestions.

Round 2

Reviewer 2 Report

I have reviewed the responses to my reviews and agree that all have been addressed.

Author Response

We appreciate the wise and thoughtful comments made by the Editor and Reviewers, which have helped to improve our manuscript.

Reviewer 3 Report

I strongly suggest the authors to send their manuscript to the journal English editing services for English review (grammar and syntax). In addition, the authors should revise the ‘instruction for authors’ section before submitting the manuscript to make sure that the submission fulfills all the requirements of this journal (see point 1, 2 and 3).

Please, make sure when the track changes of this manuscript are approved that the results of the study are matched in both the text and the tables.

1) According to the instruction for authors. The reference must be numbered in order of appearance in the text. Please amend (e.g: reference 5 and 7 in line 49. Reference 6, in line 56).

2) According to the instruction for authors. In the text, reference numbers should be placed in square brackets [ ], and placed before the punctuation (e.g:  line 47, control trials. [1-4]).

3) According to the instruction for authors. The resolution of the figures is poor. Please provide higher resolution of the figures (minimum 1000 pixels width/height, or a resolution of 300 dpi or higher).

4) Line 49, grammar, verb tense is missing: ‘although different between periods’. Replace by, ‘although difference between periods was observed,’.

5) Index, find a synonym or use it appropriately

6) Line 54. We and others…. However, only one reference is stated.

7) Line 18: remove second bracket after USA

8) Line 51: what studies? Please cite.

9) From line 256 to line 261, it appears as if the authors are describing a figure legend rather than a result. Please, amend this paragraph and present the data with the same format as in page 8. (e.g.: the figure reference in brackets rather than starting the sentence, the blue and red dashed line description should be presented in the figure legend).

10) Discussion. Line 315. Redundancy: Previously and previous is presented in one sentence.

Author Response

I strongly suggest the authors to send their manuscript to the journal English editing services for English review (grammar and syntax). In addition, the authors should revise the ‘instruction for authors’ section before submitting the manuscript to make sure that the submission fulfills all the requirements of this journal (see point 1, 2 and 3).

Authors’ Answer: We thank the Reviewer for taking the time to evaluate our manuscript. English grammar and syntax have been revised by a native English speaker. As suggested, we have checked author’s guidelines and have amended the text accordingly.

Please, make sure when the track changes of this manuscript are approved that the results of the study are matched in both the text and the tables.

Authors’ Answer: We thank the Reviewer for this comment. As suggested, we have checked and amended the text.

1) According to the instruction for authors. The reference must be numbered in order of appearance in the text. Please amend (e.g: reference 5 and 7 in line 49. Reference 6, in line 56).

Authors’ Answer: We thank the Reviewer for this comment. As suggested, we have checked and amended the numbering of references.

2) According to the instruction for authors. In the text, reference numbers should be placed in square brackets [ ], and placed before the punctuation (e.g:  line 47, control trials. [1-4]).

Authors’ Answer: We thank the Reviewer for this comment. As suggested, we have checked and amended the format of references.

3) According to the instruction for authors. The resolution of the figures is poor. Please provide higher resolution of the figures (minimum 1000 pixels width/height, or a resolution of 300 dpi or higher).

Authors’ Answer: We thank the Reviewer for this comment. The original figures were in high resolution. The inclusion of the figures in the text has reduced their quality. However, the figures have been also uploaded separately.

4) Line 49, grammar, verb tense is missing: ‘although different between periods’. Replace by, ‘although difference between periods was observed,’.

Authors’ Answer: We thank the Reviewer for this comment. As suggested, we have amended the text accordingly.

5) Index, find a synonym or use it appropriately

Authors’ Answer: We thank the Reviewer for this comment. We have amended the sentence as follows “we reported that, although difference between periods was observed, the major predictors and mechanisms of events up to 3 years after BRS implantation were device and procedural parameters.” Using “BRS implantation” instead of “index”.

6) Line 54. We and others…. However, only one reference is stated.

Authors’ Answer: We thank the Reviewer for this comment. As suggested, we have amended the text as follows: “We and others previously described how the mechanical/physical characteristics of first-generation scaffolds may explain these associations [6,7].”

7) Line 18: remove second bracket after USA

Authors’ Answer: We thank the Reviewer for this comment. As suggested, we have amended the text accordingly.

8) Line 51: what studies? Please cite.

Authors’ Answer: We thank the Reviewer for this comment. As suggested, we have amended the text citing the references of the studies.

9) From line 256 to line 261, it appears as if the authors are describing a figure legend rather than a result. Please, amend this paragraph and present the data with the same format as in page 8. (e.g.: the figure reference in brackets rather than starting the sentence, the blue and red dashed line description should be presented in the figure legend).

Authors’ Answer: We thank the Reviewer for this comment. As suggested, we have amended the text accordingly.

10) Discussion. Line 315. Redundancy: Previously and previous is presented in one sentence.

Authors’ Answer: We thank the Reviewer for this comment. As suggested, we have amended the text accordingly.

This manuscript is a resubmission of an earlier submission. The following is a list of the peer review reports and author responses from that submission.

Round 1

Reviewer 1 Report

I would like to thank the authors for this interesting submission on the long term follow-up of BRS scaffold implantation. This adds important results to current literature. However, I have some technical reservations which need to be addressed before publication:

Major comments:

1)How was normality assessed? did you use any statistical method? Please explain.

2) Multivariate Cox regression is necessary. Univariate analysis is not appropriate in your case where several risk factors are assessed. Please provide such an analysis.

3) What is the need for the sentence "all analysis should be considered exploratory"

4) KM survival analysis should be presented as one analysis without separating 1st 2nd-3rd and 4th-5th. The analysis already presented covers information about the total survival of the grafts over the whole 5 year period. Please provide unified survival graphs and add the rest to the supplementary 

5) Fig 1 and 2 need to be modified according to comment 4 and  be merged

Author Response

Thank you very much for your comments. We have tried to adress them all in a point-by-point list, which you will find below. We hope you will find our responses satisfactory.

1)How was normality assessed? did you use any statistical method? Please explain.

Thank you very much for this comment. We apologize for this omission. We changed the statistical paragraph as follows: Continuous data are described as mean and standard deviation or median and interquartile range and were compared using a parametric or nonparametric tests based on the inspection of the Q-Q plots.

This is a standard method given the shortcomings of traditional normality tests. 

2) Multivariate Cox regression is necessary. Univariate analysis is not appropriate in your case where several risk factors are assessed. Please provide such an analysis.

You are absolutely right, sorry for not pointing this out clearly. We did perform multivariate Cox analysis on the 4-5 years outcomes (we previously showed that procedural variables are the major detemirnants of procedural outcomes within 1 year and in the interval 2-3 years after implantation). 

This is now clearly described in the methods: 

Uni- and multivariate Cox regression analysis was used to describe the association between clinical/procedural parameters and outcome events.

and in the results section: 

In multivariate Cox regression model incorporating both eGFR and diabetes, the only independent predictor associated with TLF during the 4-5 years interval was diabetes (HR 6.21[1.99-19.40], P=0.002) (Supplemental Table 4).

3) What is the need for the sentence "all analysis should be considered exploratory"

Since we perform a number of different analyses, there is a risk that one of statistical tests may result "positive" (as usually defined for a P<0.05) by chance. We used this expression to make the point that Ps are presented without using the expression "statistically significant". Of course this sentence can be deleted if the reviewer believes so. 

4) KM survival analysis should be presented as one analysis without separating 1st 2nd-3rd and 4th-5th. The analysis already presented covers information about the total survival of the grafts over the whole 5 year period. Please provide unified survival graphs and add the rest to the supplementary 

5) Fig 1 and 2 need to be modified according to comment 4 and  be merged

Thank you for these comments. Your point is well taken. As suggested, we moved Figure 1 to the supplement and introduced "overall" KM curves in the manuscript. 

As of Figure 2, if reviewer agrees, we would like to do the opposite (time-dependent KM curves in the main paper and overall KM curves in the supplement). The scope of this figure is exactely to describe the impact of each of the three variables during the three time intervals. This is more clearly explained with 9 separate KM curves, even though, we admit, the figure is quite busy. 

Reviewer 2 Report

This is article showing data from the prospective observational study - 512  patients treated with coronary bioresorbable scaffolds (BRS).    The aim of the study was  evaluation on the long-term outcomes after BRS implantation in an all-comer cohort of patients with coronary artery disease.

Advantages:

The authors present a relatively large population – 512 patients treated with coronary bioresorbable scaffolds (BRS).   The follow-up lasted 5 years

Issues:

Remarks:

1.      There is no information in Supplemental Table 1 regarding the eGFR and LVEF values (mean or median).

2.      No explanation of abbreviations used in the tables.

3.      No explanation ATM in a line 112.

4.      Figures 1 and 2 have no a-c markings.

5.      In the footnote of Figure 2, the description applies only to the first analysis.

6.      In a line 190 was written “Supplemental Table 3 shows the associations of the pre-specified predictors of univariate regression analysis for the time intervals of one year, 2-3 years and 4-5 years, respectively” but there is only “Supplemental Table 3. Predictors of TLF in univariate cox regression analysis 4-5 years after implantation” – No analysis for one year and 2-3 years.

7.      Line 216 “In univariate Cox regression analysis, including all variables listed in Supplemental Table 1, only the classic cardiovascular risk  factors diabetes (HR 7.50[2.47-22.82], P=0.0004) and impaired renal function (HR 0.97[0.95-0.99], P=0.02) were predictors for the incidence of TLF (Supplemental Table 3). In multivariate Cox regression model incorporating both eGFR and diabetes, the only independent predictor remaining significant is diabetes (HR 6.21[1.99-19.40], P=0.002) (Supplemental Table 4).”  Does the analysis refer only 4-5 years after the intervention? This is not clear from the text.

This study does not contain practical aspects. This study adds nothing new. Previous studies, among others Gori T. [JACC Cardiovasc Interv. 2017 Dec 11; 10 (23): 2363-2371] showed that the technique used at the time of the implantation as well as, in particular, a meticulous attention to vessel size, BRS sizing, and device or vessel expansion were strong predictors of coronary scaffold thrombosis (ScT).

Author Response

Thank you very much for your comments. Please find below our responses, which we hope you will find appropriate. 

1.      There is no information in Supplemental Table 1 regarding the eGFR and LVEF values (mean or median).

Thank you very much for this comment. The information is now available: 

eGFR, (ml/min)

83 (69-99.5)

LVEF (%)

55 (50-55)

2.      No explanation of abbreviations used in the tables.

We apologize. All abbreviations are now detailed. 

3.      No explanation ATM in a line 112.

Thank you, this is now corrected. 

4.      Figures 1 and 2 have no a-c markings.

Sorry, this has been corrected. 

5.      In the footnote of Figure 2, the description applies only to the first analysis.

We apologize. This has been changed. 

6.      In a line 190 was written “Supplemental Table 3 shows the associations of the pre-specified predictors of univariate regression analysis for the time intervals of one year, 2-3 years and 4-5 years, respectively” but there is only “Supplemental Table 3. Predictors of TLF in univariate cox regression analysis 4-5 years after implantation” – No analysis for one year and 2-3 years.

Sorry. This was poorly explained in our first version. We revised paragraph 3.5 and 3.6 to correct this mistake. 

7.      Line 216 “In univariate Cox regression analysis, including all variables listed in Supplemental Table 1, only the classic cardiovascular risk  factors diabetes (HR 7.50[2.47-22.82], P=0.0004) and impaired renal function (HR 0.97[0.95-0.99], P=0.02) were predictors for the incidence of TLF (Supplemental Table 3). In multivariate Cox regression model incorporating both eGFR and diabetes, the only independent predictor remaining significant is diabetes (HR 6.21[1.99-19.40], P=0.002) (Supplemental Table 4).”  Does the analysis refer only 4-5 years after the intervention? This is not clear from the text.

Sorry, you are right. This has been corrected. 

This study does not contain practical aspects. This study adds nothing new. Previous studies, among others Gori T. [JACC Cardiovasc Interv. 2017 Dec 11; 10 (23): 2363-2371] showed that the technique used at the time of the implantation as well as, in particular, a meticulous attention to vessel size, BRS sizing, and device or vessel expansion were strong predictors of coronary scaffold thrombosis (ScT). 

We respectfully disagree. We previously did report that procedural variables are the major determinants of events during early and late (up to 3 years) follow-up. We now report that this association is lost beyond 3 years, ie when the device is expected to be resorbed. This data is important, as it provides information that might be valuable also for other types of scaffolds (once the device is resorbed, differences among devides would be expected not to play a role). Also, the current data provide evidence of safety after device resorption. This information is critical for the concept of BRS, including those of future new generations.